# FaSAS: A Feedback-Augmented Stepwise Algorithm Selection for Software Verification

## Abstract

Appropriate algorithm selection is a critical challenge in software verification, which typically demands domain expertise and non-trivial manpower. However, existing selectors, either dependent on machine-learned strategies or manually crafted heuristics, encounter issues such as reliance on high-quality samples with ground truth algorithm labels and limited scalability. In this paper, we propose an automated algorithm selection approach, `FaSAS`, for software verification. `FaSAS` embeds the code property graph of a semantic-preserving transformed program to enhance the robustness of the prediction model. Furthermore, our approach decomposes the selection task into the sub-tasks of predicting potentially applicable algorithms and matching the most appropriate verifiers. It further incorporates a feedback mechanism to refine predictions iteratively. Experimental results demonstrate the effectiveness of `FaSAS`, achieving a prediction accuracy of 91.47% without ground truth algorithm labels provided during the training phase. Moreover, `FaSAS` exhibits the least resource overhead compared to other approaches while solving the most verification tasks.

## 1 Introduction

Today, many verification techniques (Beyer, 2024) with varying performance capabilities have emerged to ensure important properties of software. This diversity, however, poses a challenge for software developers in algorithm selection (Rice, 1976): Which technique is most suitable for the program verification task at hand? Answering this question usually requires domain expertise. Furthermore, in the industry with immense demand for software verification (Efremov et al., 2018; Dordowsky, 2015; Blanchard et al., 2015), manually selecting plausible algorithms remains the primary approach, requiring non-trivial manpower.

In order to reduce human effort, some automated algorithm selection techniques based on designed heuristics have been introduced (Baier et al., 2024; Darke et al., 2021). These approaches select the most suitable verification algorithm from a pre-defined set of algorithms by taking into account the information such as the syntax, semantics, and properties of the verification inputs. VeriAbs (Darke et al., 2021) classifies programs into only four types based on the code structure and interval analysis results and employs corresponding algorithms, such as $k$-Induction (Donaldson et al., 2011), for verification. Several studies, *e.g.*, CPAchecker (Baier et al., 2024) and PIChecker (Su et al., 2023), have further refined program categories and designed composite strategies for them. These strategies, which sequentially combine multiple verification algorithms, have proven to be more effective than each algorithm alone. CFStra (Su et al., 2024) further combines a predefined selector with Large Language Models (LLMs), using LLMs to identify code features and then automatically selecting a verification strategy based on the identified features. However, the above heuristic approaches heavily rely on expert experience and typically select verification algorithms according to a certain program feature, making it difficult to ensure selection accuracy.

In contrast, machine learning-based selection approaches (Richter et al., 2020; Richter & Wehrheim, 2021; Leeson & Dwyer, 2024) train mapping functions from program features to verification algorithms on a large number of samples, typically showing high selection accuracy. CST (Richter & Wehrheim, 2021) uses the *abstract syntax tree* (AST) to represent programs and leverages attention mechanisms (Vaswani et al., 2017) to learn which parts of the AST are relevant to algorithm performance. The two graph-based approaches, WLJ (Richter et al., 2020) and Graves (Leeson &

Dwyer, 2024), enrich program representations by integrating control flow and data dependencies utilized by verification algorithms during the analysis process, thereby making the performance differences between algorithms more apparent. However, most of these methods require high-quality labeled datasets with algorithm tags during training, which are usually hard to collect. In addition, the prediction accuracy of these approaches decreases with program modifications or the integration of new verifiers, revealing their weaknesses in robustness and scalability.

In this paper, an automated algorithm selection approach, namely `FaSAS` is proposed for software verification. Our approach is based on the observation that verifiers producing correct verification results typically implement certain appropriate algorithms, and the supported algorithms by these verifiers indirectly reflect which ones are potentially applicable for current verification tasks. Thereby, `FaSAS` does not require the ground truth algorithm-labeled datasets during the training phase. Specifically, `FaSAS` utilizes state-of-the-art *graph neural network* (GNN) (Scarselli et al., 2009; Liu et al., 2024b) to embed the *code property graph* (CPG) of the semantic-preserving transformed verification program, effectively improving the robustness of the prediction model. Further, our approach decomposes the selection task into the sub-tasks of predicting potentially applicable algorithms and matching the most appropriate verifiers. The verifiers that implement the potentially applicable algorithms may also succeed in the same verification tasks. This stepwise prediction method allows for only retraining the matching model when introducing new verifiers, thereby increasing the scalability of `FaSAS`. Additionally, our approach also introduces a feedback loop on incorrect predictions, reducing the manual effort required to adjust verification algorithms while improving the final prediction accuracy.

Experiments have been carried out on the benchmarks of SV-COMP (Beyer, 2024). We evaluate `FaSAS` on 20 verifiers and over 15,000 verification tasks. Experimental results demonstrate the effectiveness of our approach, achieving a prediction accuracy of 91.47%. Moreover, compared with other selectors, our approach requires the least resource overhead while solving the most verification tasks.

Our contributions are summarized as follows:

1) *Novelty*. We present an effective, robust, and scalable approach for automatically selecting the most suitable verification algorithm without high-quality labeled samples.

2) *Practical Approach*. We address the challenges of algorithm selection by incorporating graph neural network embeddings, multi-model stepwise predictions, and feedback adjustment mechanisms.

3) *Open Source*. We have developed and implemented our approach as a tool named `FaSAS`. We have made the implementation, along with all relevant publicly available data, accessible to facilitate comparison: https://figshare.com/s/746cb529fab12742644c.

4) *Evaluation*. We conducted an extensive comparison against multiple SOTA algorithm selection approaches (e.g., Graves, CST, and CFStra) across a diverse set of verification tasks to demonstrate the effectiveness, robustness, and scalability of our approach.

## 2 PRELIMINARIES

### 2.1 GRAPH-BASED CODE REPRESENTATION AND LEARNING

In most algorithm selectors, whether based on machine learning or manually designed heuristics, program features serve as the determining factors in selecting an appropriate verification algorithm. These features are usually derived from the program's graph-based representations such as *abstract syntax tree* (AST), *control flow graph* (CFG), *program dependency graph* (PDG), and *code property graph* (CPG). In this paper, we focus on the CPG that provides comprehensive program information.

**Code Property Graph (Yamaguchi et al., 2014).** For a given program $P$, the code property graph $\mathbb{G}_P = (\mathbb{V}, \mathbb{E}, \lambda, \mu)$ is a directed, edge-labeled, attributed multigraph that is constructed from the AST, CFG and PDG of $P$, where $\mathbb{V}$ is a set of AST nodes, $\mathbb{E} \subseteq (\mathbb{V} \times \mathbb{V})$ is a set of directed edges. $\lambda : \mathbb{E} \to \Sigma$ is an edge labeling function assigning a label from the alphabet $\Sigma$ to each edge. Properties can be assigned to edges and nodes by the function $\mu : (\mathbb{V} \cup \mathbb{E}) \to 2^S$ where $S$ is the set of properties.

Within a CPG, all nodes in the CFG and PDG can be represented by nodes in the AST, while the edges from the three graphs collectively form the connections in the CPG. Intuitively, CPGs offer a

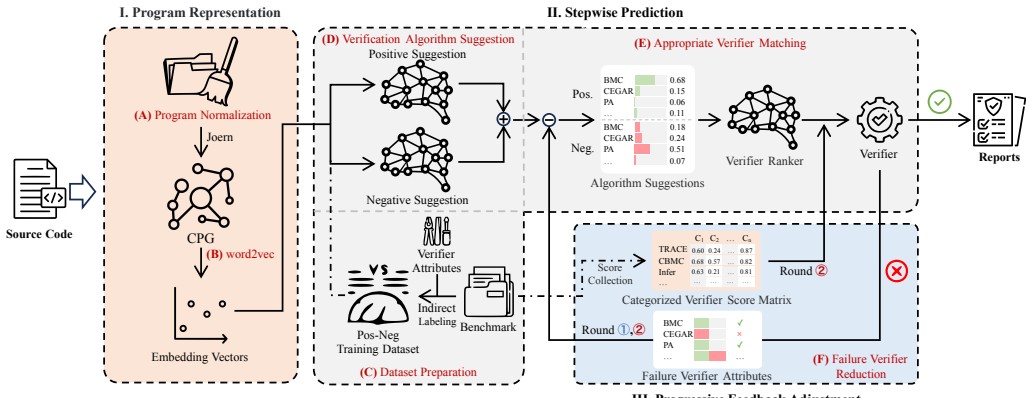

Figure 1: Overview of `FaSAS`

more comprehensive set of program features compared to ASTs, CFGs, and PDGs, making them well-suited for code representation. For example, Liu et al. (Liu et al., 2023) employ a CPG-based neural network to learn representations of CPGs through iterative information propagation, and the experiments demonstrate strong performance in code clone detection and code classification tasks.

**Graph-based Learning Approaches.** Graph neural networks (Liu et al., 2024b) can effectively handle the data with relationships and dependencies that need to be represented in graphs. In addition to capturing node features within a graph, GNNs utilize a mechanism called "message passing" to aggregate the features of each node with those from its neighbors. This iterative refinement allows effective modeling of dependencies between nodes. Gated graph neural networks (GGNNs) (Beck et al., 2018), graph convolutional neural networks (GCNN) (Zhang et al., 2019), and graph attention networks (GATs) (Veličković et al., 2017) are all graph neural networks that integrate different techniques. For example, GATs leverage masked self-attention layers to address the shortcomings of distinguishing node importance in traditional GNNs, enabling the model to learn to focus more on relevant nodes while diminishing attention on distant ones. Due to their exceptional performance, GNNs have been widely applied across various domains, including text classification, vulnerability detection, and recommendation systems (Ma et al., 2021; Nguyen et al., 2022; Chen et al., 2024).

## 2.2 ALGORITHM SELECTION PROBLEM

Existing verifiers can support multiple verification algorithms, while algorithms based on different theories may be applicable to verify the same program. The goal of algorithm selection is to choose the optimal one from the existing verifiers (*i.e.*, achieving correct verification results while minimizing runtime and memory overheads), which requires an in-depth knowledge of these algorithms.

**Algorithm Selection (Leeson & Dwyer, 2024).** Given a program $P \in \mathcal{P}$, a correctness specification $\phi \in \Phi$, and a suite of verifiers $\mathcal{V}$ which can accept $P$ and $\phi$, rank $\mathcal{V}$ according to both their ability to correctly determine the truth of whether $P$ satisfies $\phi$ and their efficiency in terms of resource consumption, with the former being prioritized over the latter.

This requires that both the program $P$ and specification $\phi$ can be accepted by any verifier $v \in \mathcal{V}$. For simplicity, we assume the specification to verify, typically assertion-based (Clarke & Rosenblum, 2006), is encoded inside the program. Each verifier maps the program $P$ and the specification $\phi$ to a verification outcome $\{T, F, U\}$ ($T$ = program is correct, $F$ = program contains error, $U$ = unknown), with verifiers outputting $U$ only in case of violating resource constraints, such as timeout or memory exhaustion.

## 3 FASAS

Fig. 1 shows an overview of `FaSAS`. The workflow comprises three main steps: ❶ **Program Representation** (Sect. 3.1). The input program is normalized through a semantic-preserving transformation, after which `FaSAS` converts the transformed program into code property graphs and utilizes word embedding based on GNN to initialize node and edge embedding vectors. This step

aims to reduce the differences between programs with similar structures, and provides rich syntax, semantic, and structural information for subsequent steps. ❷ **Stepwise Prediction** (Sect. 3.2). `FaSAS` uses the attribute information of multiple verifiers to indirectly label the dataset consisting solely of verification inputs and outcomes. Subsequently, two models trained on the labeled dataset provide positive and negative suggestions for selecting verification algorithms, respectively. Thereafter, `FaSAS` matches the verifier whose attributes are closest to the suggested algorithm combinations, and the selected verifier will be applied to the verification task. ❸ **Progressive Feedback Adjustment** (Sect. 3.3). In cases of verification failure or other scenarios requiring reselection of the verifier, `FaSAS` conducts two rounds of progressive feedback adjustments for the new verifier based on the matched results from the previous steps. Specifically, in the round ①, `FaSAS` only subtracts the attributes of the failed verifier proportionally from the suggested algorithm combinations predicted in step 2 to mitigate their impact on the prediction, and then re-matches the new verifier. In the round ②, in addition to reducing the impact of failed verifiers and matching multiple appropriate verifiers, `FaSAS` also selects the highest-scoring unselected verifier based on the performance of these matched verifiers on the dataset. We will explain the methodology of each step in detail.

## 3.1 PROGRAM REPRESENTATION

For the input program to be verified, the goal of this step is to generate vector representations that encompass rich features while minimizing their impact on the robustness of prediction models.

Algorithm 1 illustrates the details of achieving this goal. Firstly, as depicted in Fig. 1 (A), `FaSAS` normalizes the given program $P$ by applying a semantic preserving transformation. This transformation uniquely maps each variable and function name in $P$ to a fixed vocabulary without modifying its structure or introducing new semantics. This simple program

---

**Algorithm 1:** Program Representation

**Input:** The program $P \in \mathcal{P}$ to be represented.
**Output:** The set $\mathbb{X}$ of initial node embedding vectors of $P$.

1   $P' \leftarrow \text{Normalize(P)}$      ▷ *program normalization*
2   $\mathbb{G}_{P'}(\mathbb{V}, \mathbb{E}, \lambda, \mu) \leftarrow \text{Joern}(P')$    ▷ *construct the CPG for $P'$*
3   $\mathcal{X}_P \leftarrow \emptyset$
4   **for** $n \in \mathbb{V}$ **do**
5      $\mathbb{E}_n \leftarrow \{e | n \in e, e \in \mathbb{E}\}$    ▷ *obtain the related edges of $n$*
6      $\mathbf{l}_{in}^n \leftarrow \max(\{\text{word2vec}(\lambda(e)) | e \in \mathbb{E}_n, n = \text{dst}(e)\})$
7      $\mathbf{l}_{out}^n \leftarrow \max(\{\text{word2vec}(\lambda(e)) | e \in \mathbb{E}_n, n = \text{src}(e)\})$
8      $\mathbf{l}_n \leftarrow \mathbf{l}_{out}^n + \mathbf{l}_{in}^n$      ▷ *edge embedding*
9      $\mathbf{p}_n \leftarrow \text{type}(n) \parallel \text{word2vec}(\mu(n))$   ▷ *property embedding*
10      $\mathbf{x}_n = \mathbf{l}_n \parallel \mathbf{p}_n$      ▷ *node embedding*
11      $\mathcal{X}_P \leftarrow \mathcal{X}_P \cup \{\mathbf{x}_n\}$
12   **return** $\mathcal{X}_P$

---

normalization helps reduce noisy data caused by personalized naming and minimizes differences between similarly structured programs, thereby enhancing the robustness of prediction models to minor changes in program contents. Then, `FaSAS` utilizes Joern to construct the code property graph $\mathbb{G}_{P'} = (\mathbb{V}, \mathbb{E}, \lambda, \mu)$ of the semantic-preserving transformed program $P'$. Thereafter, the set $\mathcal{X}_P \in \mathbb{X}$ of initial embedding vectors for the nodes $n \in \mathbb{V}$ will be computed, where $\mathbb{X} = \{\mathcal{X}_P | P \in \mathcal{P}\}$ represents the initial embedding vector set of all programs $P \in \mathcal{P}$. Specifically, the loop begins by retrieving the incoming and outgoing edges of $n$ (line 5). Subsequently, the algorithm aggregates the larger components of the label embedding vectors of incoming and outgoing edges to obtain the edge embedding $\mathbf{l}_n$ (lines 6-8). Here, $\text{src}(\cdot)$ and $\text{dst}(\cdot)$ are functions to obtain the source and destination nodes, and $\text{word2vec}(\cdot)$ is a pre-trained model utilized to embed the word vector for the sentence element in $\mathbb{G}_{P'}$. After that, the property embedding $\mathbf{p}_n$ is obtained by concatenating the node type with the embedding vector of node property $\mu(n)$ (line 9). Finally, the initial node embedding vector of $n$ can be obtained by concatenating the two parts (line 10).

For the approaches that do not apply the semantic preserving transformations, e.g., `CST` (Richter & Wehrheim, 2021), modifications in program variables could result in decreased robustness in predictions. In contrast, the normalization sub-step reduces the sensitivity of our prediction model to program modifications, thereby enhancing the robustness of the prediction output. In addition, different from encoding edge types into initial node vectors (Zhang et al., 2023), our method enhances the representation of nodes within the CPG by incorporating additional label information from neighboring edges for each node, thereby increasing the distinctiveness among different nodes.

## 3.2 STEPWISE PREDICTION

This step includes two critical sub-steps: 1) predicting positive and negative algorithm suggestions for software verification (Fig. 1 (D)), and 2) matching these suggestions with the most appropriate verifier (Fig. 1 (E)). Before performing these two sub-steps, it is essential to prepare training datasets for the first sub-step by leveraging the motivating observation (Fig. 1 (C)).

### 3.2.1 DATASET PREPARATION

Algorithm-labeled high-quality datasets can significantly improve the accuracy of a prediction model. However, manually collecting such datasets is generally a challenge due to the implicit verifier invocation parameters and complex outputs. In order to establish implicit connections between verification outcomes and potentially applicable algorithms, we indirectly annotated the algorithms applicable to each verification task. Specifically, for a given verification task, the suggestion weights (*i.e.*, annotated labels) of algorithms for verifying this task can be calculated proportionally using the verification scores according to the algorithms supported by multiple verifiers.

Formally, we define the labeling function $L : \mathcal{P} \to \mathcal{A}$. Here, $\mathcal{P}$ represents a set of programs, $\mathcal{A} \in \mathbb{R}^m$ denotes the annotated set of $m$-dim suggestion weight of algorithms in real number, and $m$ is the total number of algorithms supported by the verifiers in $\mathcal{V}$. Each verifier $v \in \mathcal{V}$ supports a fixed set of algorithms, and the algorithm implementation matrix $\mathbf{M} \subseteq \mathbb{R}^{n \times m}$ of $n = |\mathcal{V}|$ verifiers can be directly obtained from their documentation by checking whether $v$ implements the given set of algorithms. The suggestion weights of algorithms for a program $P$ can be calculated as:

$$L(P) = \mathbf{M}^T \mathbf{s}_P \qquad (1)$$

Here, $\mathbf{s}_P \subseteq \mathbb{R}^n$ represents the scores of $n$ verifiers $v_i \in \mathcal{V}(1 \le i \le n)$ for $P$, each score $s_P^i \in \mathbf{s}_P(1 \le i \le n)$ is further calculated based on the verification outcome $r_o^i \in \{TP : 2, FP : -2, TN : 2, FN : -1, U : 0\}$[1], the time consumption $r_t^i$, and the memory usage $r_m^i$:

$$s_P^i = r_o^i - \frac{r_t^i}{T_{limit}} - \frac{r_m^i}{M_{limit}} \qquad (2)$$

, where $T_{limit}$ and $M_{limit}$ represent the upper limits of verification resources.

Intuitively, Eq. (2) balances the verification accuracy with the performance of the verifier. The term $r_o^i$ increases the penalty for the verifiers that fail to detect specification violation (*i.e.*, false positive). Moreover, the sign of $s_P^i$ indicates the positive or negative impact of each verifier on the suggestion weights of the algorithm. Therefore, Eq. (1) provides algorithm suggestions for verifying program $P$ by synthesizing the performance from different verifiers.

To avoid the cancellation of positive and negative influences that might hinder a model from learning effective algorithm suggestion patterns, we construct separately labeled datasets $\mathbb{D}^+, \mathbb{D}^- \subseteq \mathcal{P} \times \Phi \times \mathcal{A}$ with positive and negative labels $L^+(P) = \mathbf{M}^T \mathbf{s}_P^+$ and $L^-(P) = \mathbf{M}^T \mathbf{s}_P^-$ for each program $P \in \mathcal{P}$, where $\Phi$ denotes a set of correctness specifications, $\mathbf{s}_P^+$ and $\mathbf{s}_P^-$ mask the influences of verifiers with negative and positive scores in $\mathbf{s}_P$ by setting $s_P^i(1 \le i \le n)$ to 0, respectively.

### 3.2.2 VERIFICATION ALGORITHMS SUGGESTION

In this sub-step, FaSAS utilizes the indirectly labeled datasets, $\mathbb{D}^+$ and $\mathbb{D}^-$, with the initial embedding vector set $\mathbb{X}$ of all programs $P \in \mathcal{P}$ to train the positive and negative suggestion models, $\mathcal{S}^+, \mathcal{S}^-$ : $\mathbb{X} \to \mathcal{A}$. Since both models are trained in the same manner, we only showcase the training approach of one of the models.

Firstly, for a program $P \in \mathcal{P}$, FaSAS employs a GNN comprising a GraphSAGE (Hamilton et al., 2017; Liu et al., 2024a) layer and a global max pooling layer to extract graph features from $\mathbb{G}_P$. In the GraphSAGE layer, the embedding vectors $\mathbf{x}_i \in \mathcal{X}_P(1 \le i \le |\mathcal{X}_P|)$ of each node in $\mathbb{G}_P$ can be updated by using:

$$\mathbf{x}_i' = \mathbf{W}_1 \mathbf{x}_i + \mathbf{W}_2 max_{j \in \mathcal{N}(i)} \mathbf{x}_j \qquad (3)$$

---

[1]$TP$ stands for true positive, $FP$ stands for false positive, $TN$ represents true negative, $FN$ denotes false negative, and $U$ represents unknown.

, where $\mathbf{W}_1$ and $\mathbf{W}_2$ are weight matrices to be learned, $\mathcal{N}(i)$ denotes the number of neighbors of node $\mathbf{x}_i$. Intuitively, this layer aggregates information from the neighboring nodes $\mathbf{x}_j$ of the current node $\mathbf{x}_i$ by using the $max$ aggregation function, and combines the results with the information of node $\mathbf{x}_i$ itself to update its feature representation $\mathbf{x}_i'$. Compared to the methods based on transductive learning frameworks like DeepWalk (Perozzi et al., 2014; Harker & Bhaskara, 2024), GraphSAGE, based on inductive learning, can learn from unseen nodes, making it effective in extracting implicit information from complex programs. In addition, this optimization adapts full graph sampling in *graph convolutional network* (GCNs) (Zhang et al., 2019; Myung et al., 2024) to neighbor sampling centered on individual nodes, enabling distributed training of large-scale graphs.

In the global max pooling layer, the feature vectors of all nodes in the original $\mathbb{G}_P$ and the updated by GraphSAGE layer are aggregated into a graph feature vector $\mathbf{x}_P$, and $\mathbf{x}_P$ is expressed as:

$$\mathbf{x}_P = max_{i=1}^{|\mathcal{X}_P|}\mathbf{x}_i \parallel max_{i=1}^{|\mathcal{X}_P|}\mathbf{x}_i' \tag{4}$$

This layer is designed to obtain a fixed-size graph feature representation of $\mathbb{G}_P$ for performing the prediction task of algorithm suggestion. The operator $\parallel$ concatenates the original features of graph $\mathbb{G}_P$ with the aggregated features, allowing the subsequent prediction network to effectively capture algorithm suggestion relationships from both global and local program features.

Thereafter, `FaSAS` uses a neural network with two convolution layers and two max pooling layers to perform the prediction task of algorithm suggestion. This network produces a score vector $\mathbf{a}_P \in \mathbb{R}^m$ for the considered $m$ types of algorithms. We combine a Sigmoid layer and the BCELoss (*i.e.*, BCEWithLogitsLoss) as the prediction loss function, so the loss function $\mathcal{L}_{\mathbf{y}}$ of label $\mathbf{y} = L(P)$ is expressed as:

$$\mathcal{L}_{\mathbf{y}} = \text{BCEWithLogitsLoss}(\mathbf{y}, \mathbf{a}_P^\alpha) \tag{5}$$

, where $\mathbf{a}_P^\alpha$ represents the part of $\mathbf{a}_P$ where the ratio of sorted cumulative value surpasses the threshold $\alpha$ (i.e., the other parts will be set to 0). The threshold $\alpha$ controls the algorithm suggestion model to prioritize important algorithms and ignore less relevant ones. After the training phase, the positive and negative suggestions of algorithms can be utilized to rank the $n$ verifiers $v \in \mathcal{V}$, in terms of correctness, followed by resource overheads.

### 3.2.3 APPROPRIATE VERIFIER MATCHING

In this sub-step, `FaSAS` uses the algorithm suggestions $\mathbf{a}_P^+$ and $\mathbf{a}_P^-$ predicted by the positive and negative suggestion models $\mathcal{S}^+$ and $\mathcal{S}^-$ to train the ranking model, $\mathcal{R} : \mathcal{A}' \to \mathbb{R}^{|\mathcal{V}|}$, where $\mathcal{A}' \in \mathbb{R}^{2m}$.

Specifically, `FaSAS` utilizes a simple three-layer fully connected neural network that employs the concatenated suggestions $\mathbf{a}_P^+ \parallel \mathbf{a}_P^-$ as the input data and the scores $\mathbf{s}_P$ from $n$ verifiers for $P$ as the label to train the ranker of the verifiers. The model $\mathcal{R}$ outputs the ranking vector $\mathbf{r}_P \in \mathbb{R}^{|\mathcal{V}|}$ of these verifiers. We use the *negative log-likelihood loss* (NLLLoss) combined with the *log softmax* (LogSoftmax) function as the loss function, hence the loss function $\mathcal{L}_{\mathbf{s}_P}$ of label $\mathbf{s}_P$ is expressed as:

$$\mathcal{L}_{\mathbf{s}_P} = \text{NLLLoss}(\text{LogSoftmax}(\mathbf{r}_P), \mathbf{s}_P) \tag{6}$$

After training, the verifier corresponding to the maximum value in the ranking vector $\mathbf{r}_P$ will be prioritized for selection.

Intuitively, when new verifiers become available, unless new algorithms are introduced, our approach can simply retrain the lightweight ranking model $\mathcal{R}$, allowing `FaSAS` to be more easily scalable.

### 3.3 PROGRESSIVE FEEDBACK ADJUSTMENT

In cases of verification failure or other scenarios requiring reselection of the verifier, `FaSAS` provides a mechanism to progressively adjust the choice of new verifiers based on the attributes of previously failed verifiers and the overall performance of each verifier $v \in \mathcal{V}$ in the benchmark.

As depicted in Fig. 1 (F), this step involves two progressive rounds of failure verifier reduction. Each round adjusts the impact of failed verifier $v_i \in \mathcal{V}(1 \le i \le |\mathcal{V}|)$ on the suggestion vectors $\mathbf{a}_P^+, \mathbf{a}_P^-$ generated in Sect. 3.2.2 by employing the reduction function $\mathcal{F} : (\mathcal{A} \times \mathcal{A}) \times \mathcal{V} \to (\mathcal{A} \times \mathcal{A})$:

$$\mathcal{F}(\mathbf{a}_P^+, \mathbf{a}_P^-, v_i) : \begin{cases} \mathbf{a}_P^+ \mathrel{-}= \beta \cdot \mathbf{M}_i^T \\ \mathbf{a}_P^- \mathrel{+}= \beta \cdot \mathbf{M}_i^T \end{cases} \tag{7}$$

, where $\mathbf{M}_i^T \in \mathbb{R}^m$ represents the vector of algorithms supported by $v_i$, and $\beta$ denotes the adjustment coefficient.

Intuitively, this function proportionally decreases and increases the influence of the algorithms supported by the failed verifier $v_i$ on the algorithm suggestions $\mathbf{a}_P^+, \mathbf{a}_P^-$, thereby making the ranking model $\mathcal{R}$ more inclined to recommend other verifiers that support algorithms more suitable for the task.

Moreover, the performance of verifiers varies across different types of verification tasks. Therefore, for a program $P$ and a specification $\phi_j \in \Phi$ ($\mathcal{P} = \bigcup \mathcal{P}_j$, $1 \leq j \leq |\Phi|$), in addition to use function $\mathcal{F}$ to adjust the impact of failed verifier, round ② will further select the verifier with the best performance in categorized verifier score matrix $\mathbf{C} \in \mathbb{R}^{|\mathcal{V}| \times |\Phi|}$. Here, $|\Phi|$ represents the number of categories of verification specifications, $\mathbf{C}_{i,j} = \Sigma_{P \in \mathcal{P}_j} s_P^i$ denotes the overall score of verifier $v_i \in \mathcal{V}$ on category $\phi_j$.

Formally, for the algorithm suggestions $\mathbf{a}_P^+, \mathbf{a}_P^- \in \mathcal{A}$, the failed verifiers $v_1, v_2 \in \mathcal{V}$ ($v_1$ predicted before $v_2$), the newly selected verifier $v_3 \in \mathcal{V}(v_3 \neq v_1 \neq v_2)$ in the two rounds of adjustment can be defined as:

$$v_2, v_3 = \begin{cases} \max(\mathcal{R}(\mathcal{F}(\mathbf{a}_P^+, \mathbf{a}_P^-, v_1))) & ① \\ \max(\mathcal{R}(\mathcal{F}(\mathcal{F}(\mathbf{a}_P^+, \mathbf{a}_P^-, v_1), v_2))) \cap \max(\mathbf{C}_{\cdot,j}) & ② \end{cases} \tag{8}$$

, where $\mathbf{C}_{\cdot,j}$ represents the $j$-th column of $\mathbf{C}$. The equation of round ② indicates that the new selected verifier $v_3$ performs the best on the category $\phi_j$. If there are no such verifiers, we will use the predicted verifier $v_3 = \max(\mathcal{R}(\mathcal{F}(\mathcal{F}(\mathbf{a}_P^+, \mathbf{a}_P^-, v_1), v_2))$ after 2 rounds of reduction.

## 4  EVALUATION

**Datasets.** The SV-COMP[2] suite includes a large set of verification tasks, with the majority focused on verifying C programs encoded with different types of specifications. We use the 2024 SV-COMP benchmarks, and similar to the evaluation setup of `CST` and `Graves` we randomly selected 20 verifiers that competed in all four major categories: overflow, reach safety, termination and memory safety Beyer (2024), resulting in a total of 15,643 samples. To compare the performance differences between our approach and the most relevant selector `Graves`, and measure the generalization performance of learning, we randomly divided the benchmarks into training, evaluation, and test sets, which is the same as the way `Graves` prepares the dataset. Meanwhile, we also ensure the split reflects the populations of specifications. The generated splits maintain the same relative ratio for the verification problem per category as the source dataset.

**Baselines.** We compared `FaSAS` with two state-of-the-art machine learning-based verification algorithm selection approaches, namely `Graves` (Leeson & Dwyer, 2024) and `CST` Richter & Wehrheim (2021). We also evaluate two additional selectors: a greedy selector (`Greedy`) and a random selector (`Random`). The purpose of comparing `FaSAS` with `Greedy` is to understand the benefits of using the collected verifier performance data to select the most appropriate verifier. By collecting the verification performance metrics of multiple verifiers on the specified problem set, `Greedy` can automatically select the verifier with the best performance based on this metric when given a specific problem specification. The `Random` selector allows us to evaluate whether the evaluation metrics are effective. That is, the metric is seemingly not rigorous if the `Random` selector performs well.

**Evaluation Metrics.** We use two metrics to evaluate the algorithm selection performance of `FaSAS` compared to the baseline approaches: Successful Verifier Selection Accuracy and Top1 Successful Verifier Selection Accuracy. These two metrics have been extensively utilized in previous studies Richter et al. (2020); Richter & Wehrheim (2021); Leeson & Dwyer (2024). Successful Verifier Selection Accuracy measures the ability to select a verifier that successfully verifies a given program. On the other hand, Top1 Successful Verifier Selection Accuracy evaluates whether the first predicted verifier has the best performance on the given task.

### 4.1  PERFORMANCE EVALUATION

---

[2]https://gitlab.com/sosy-lab/benchmarking/sv-benchmarks.git, branch:svcomp24

We compare the performance of FaSAS against the baseline approaches using the mentioned evaluation metrics. Tab. 1 presents the results over the evaluation metrics. It clearly depicts that FaSAS outperforms Greedy and Random. It also demonstrates a significant improvement of 1.75x in this metric compared to CST. FaSAS aggregates the code and edge information into each CPG node. This feature aggregation can effectively enrich the program representations, thereby positively impacting the Top1. metric. In addition, there is a 2.51 percentage point increase compared to Graves. FaSAS achieved a probability of 91.47% in selecting a verifier that can successfully complete the given verification tasks.

Table 1: Comparisons Between Different Selectors

| Selector | Top1. | Suc. |
|---|---|---|
| **FaSAS** | **81.64%**±**0.24%** | **91.47%**±**0.20%** |
| Graves | 77.51%±1.28% | 88.96%±1.13% |
| CST | 29.64%±0.00% | 52.24%±0.00% |
| Greedy | 23.00%±0.00% | 48.41%±0.00% |
| Random | 10.26%±0.60% | 48.05%±0.52% |

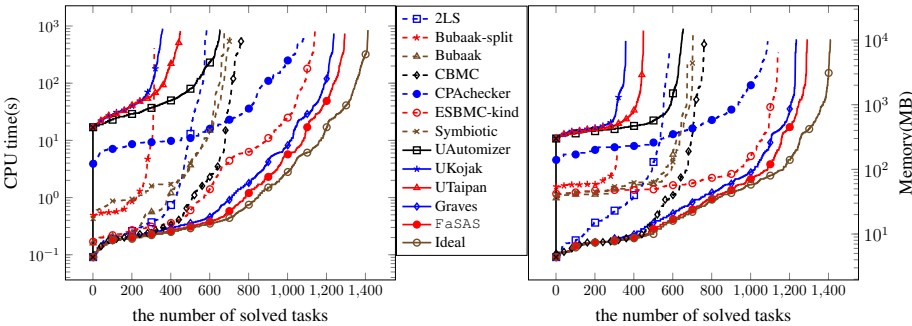

Figure 2: The quantile plots of time and memory consumptions in logarithmic scale.

Fig. 2 depicts the time and memory consumptions of a verifier or algorithm selector on the number of solved tasks, with the verification results collected from the test dataset. The curve closer to the bottom right corner in this graph indicates better performance on the test dataset. The plot clearly demonstrates that FaSAS outperforms the algorithm selector Graves and other individual verifiers (e.g., CBMC, 2LS, Symbiotic, etc.). Meanwhile, FaSAS is the closest to Ideal, which assumes that we can always select the most appropriate algorithm based on ground truth. Consequently, the superiority of FaSAS over other approaches is evident.

## 4.2 ROBUSTNESS AND SCALABILITY EVALUATION

Table 2: Comparisons for Robustness of Modification

| Selector-Status | Top1. | Suc. |
|---|---|---|
| FaSAS-Unmodified | **81.64%**±**0.24%** | **91.47%**±**0.20%** |
| Graves-Unmodified | 77.51%±1.28% | 88.96%±1.13% |
| CST-Unmodified | 29.64%±0.00% | 52.24%±0.00% |
| FaSAS-Modified | **80.62%**±**0.26%** | **90.50%**±**0.20%** |
| Graves-Modified | 73.93%±0.64% | 86.48%±0.92% |
| CST-Modified | 21.71%±0.00% | 60.88%±0.00% |

Table 3: Comparisons for Scalability of Selector

| Verifiers | Methods | Top1. | Suc. |
|---|---|---|---|
| 15 | FaSAS | **80.04%**±**0.36%** | **90.91%**±**0.45%** |
| | Graves | 72.70%±1.63% | 86.02%±1.53% |
| | Greedy | 21.80%±0.00% | 48.40%±0.00% |
| | Random | 6.28%±0.51% | 39.90%±0.90% |
| 20 | FaSAS | **78.10%**±**0.36%** | **90.63%**±**0.29%** |
| | Graves | 69.92%±1.47% | 86.04%±1.50% |
| | Greedy | 16.91%±0.00% | 48.41%±0.00% |
| | Random | 4.81%±0.52% | 36.73%±0.82% |

**Robustness.** We evaluate the robustness of FaSAS by conducting experiments under both unmodified and intentionally perturbed datasets, where the latter included minor code alterations like renaming variables and changing loop structures, etc. As shown in Tab. 2, the results revealed that FaSAS maintained high performance, with only slight decreases in success rates after modifications. This indicates that FaSAS demonstrates high robustness to minor perturbations in the input programs, which is a crucial requirement for practical deployment in software verification tasks. The robustness of FaSAS can be attributed to its use of program normalization and CPG-based program representation. This enables FaSAS to capture the fundamental structure and semantic features of programs while generalizing well across different versions of the same program when minor changes are introduced.

**Scalability.** To evaluate the scalability of FaSAS, we conducted experiments by adding 5 and 10 new verifiers, observing how performance holds up as the number of verifiers increases. The results, showcased in Tab. 3, indicate that FaSAS sustains good performance, with a minimal decline in success rates when scaling from 15 to 20 verifiers. This performance is notably better than other tools

like `Graves`, which showed more significant drops. The scalability of `FaSAS` is enhanced by its modular design, allowing efficient adaptation to new tools and continuous improvement through a feedback loop mechanism.

### 4.3 ABLATION STUDY

**Program Representation.** We compared four graph-based program representations, i.e., AST, CFG, PDG, and CPG, by training a predictive strategy model for each. The evaluation metrics include: 1) Acc: The probability that the predicted algorithm matches the required algorithms; 2) Best: The probability that the predicted algorithm suggestions exactly match the required algorithms; 3) Contain: The probability that all predicted algorithms include the required

Table 4: Comparisons for Program Representations

| Graph | Acc | Best | Contain |
|---|---|---|---|
| AST | 91.50%±0.21% | 67.37%±0.91% | 59.66%±0.42% |
| CFG | 91.38%±0.73% | 67.94%±0.39% | 55.48%±0.41% |
| PDG | 91.77%±0.29% | 66.81%±0.49% | 59.38%±0.22% |
| **CPG** | **92.25%±0.84%** | **70.35%±0.24%** | **62.07%±0.38%** |

algorithms. In the three metrics, the required algorithms refer to the positive suggested algorithms, whose sorted cumulative value surpasses the threshold $\alpha$, made by the positive suggestion model $\mathcal{S}^+$. As shown in Tab. 4, the experimental results demonstrate that the program representation based on CPG outperforms the other representations. This indicates that CPG-based representations perform better in capturing rich program features and selecting appropriate verifiers, making it the preferred choice for program representation in `FaSAS`.

**Stepwise Prediction.** To evaluate the impact of the stepwise prediction step in `FaSAS`, we compared the performance of `FaSAS` with an end-to-end version that utilizes the generated program representations to directly match the appropriate verifiers (`FaSAS-E2E`). As shown in Tab. 5, the results demonstrate

Table 5: Comparisons for Prediction Approaches

| Approach | Top1. | Suc. |
|---|---|---|
| `FaSAS-E2E` | 78.23%±0.24% | 90.39%±0.20% |
| `FaSAS` | **81.64%±0.89%** | **91.47%±0.83%** |

that this stepwise prediction approach does not decrease the accuracy of selecting appropriate verifiers. This prediction decomposition approach helps `FaSAS` focus more on improving the performance of the critical sub-steps that have the most impact on accuracy and effectiveness in the algorithm selection task.

**Feedback Adjustment Mechanism.** We also analyzed the contributions of the feedback mechanisms in improving the accuracy of model prediction. Tab. 6 shows the performance of `FaSAS` with different rounds of feedback adjustments. In the absence of feedback (i.e., None), the model achieves a `Top1 Success` rate of 81.64% and a `Success` rate of 91.47%. These ratios notably increased

Table 6: Comparisons for Feedback Adjustment Rounds

| Rounds | Top1. | Suc. |
|---|---|---|
| None | 81.64%±0.24% | 91.47%±0.20% |
| ① | 86.23%±0.29% | 97.23%±0.11% |
| ② | 87.39%±0.44% | 98.67%±0.06% |

after the round ①. Further, after the round ②, these ratios noticeably increased by 1.16% and 1.44%, respectively. These results highlight the critical role of the feedback loop in continuously improving the accuracy and effectiveness of appropriate algorithm selection.

The ablation study confirms the following key findings: (1) The CPG-based program representation approach is the most effective, surpassing other methods in capturing rich program features for selecting appropriate verifiers. (2) Stepwise prediction significantly improves performance by breaking down the selection task into sub-tasks, enabling `FaSAS` to achieve higher accuracy. (3) The feedback adjustment mechanism plays a crucial role in enhancing prediction accuracy, as each round leads to substantial improvements in both the `Top1 Success` and `Success` rates.

## 5 CONCLUSION

In this paper, we propose an automated algorithm selection approach for software verification. By incorporating graph neural network embeddings, multi-model stepwise predictions, and feedback adjustment mechanisms, our approach `FaSAS` avoids dependency on algorithm-labeled datasets while achieving high prediction accuracy and scalability. Compared with other approaches, the proposed approach achieves higher accuracy in selecting the most appropriate verification algorithm. Moreover, even with the introduction of new verifiers, `FaSAS` exhibits better scalability and robustness.

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

# A APPENDIX

## A.1 RELATED WORK

**Manually Designed Algorithm Selection** Verification techniques with varying performance capabilities pose a challenge for developers in selecting the most appropriate one. In order to reduce human effort, some automated algorithm selection approaches based on manually designed heuristics have been introduced (Baier et al., 2024; Darke et al., 2021; Su et al., 2023). VeriAbs Darke et al. (2021) classifies programs into four types and selects a suitable strategy from a pre-defined set of strategies by taking into account the syntax and semantics of the program to be verified. CPAchecker (Baier et al., 2024) and PIChecker (Su et al., 2023) have further refined program categories and designed composite strategies for them. These strategies have been proven to be more effective than using each algorithm alone. The follow-up work, PeSCo (Richter & Wehrheim, 2019), optimizes the algorithm execution order within composite strategies, effectively reducing runtime by skipping algorithms that do not fit the given verification task. However, these methods heavily rely on expert experience. They are limited in applicability scope or scalability. Compared to these methods, our approach learns the mappings from program representations to suitable verification algorithms, making algorithm selection more flexible and scalable.

**Machine Learning-based Algorithm Selection** Compared to the above methods, machine learning-based selection approaches can adapt to specific selection tasks by constructing a decision model from available data and generally exhibit high selection accuracy (Richter et al., 2020; Richter & Wehrheim, 2021; Leeson & Dwyer, 2024). CST (Richter & Wehrheim, 2021) employs representation learning to avoid the handpicking of program features and combines an attention mechanism to improve the interpretability of the learned selectors. The most related work to ours is given in Leeson & Dwyer (2024), which employs an extended AST to represent the program to be verified and utilizes a neural network to directly predict the ranking of verifiers. In contrast, FaSAS employs CPG containing richer information to represent programs and performs stepwise prediction to indirectly select suitable verifiers, thereby effectively increasing the scalability and robustness of our approach. Furthermore, unlike other algorithm selection approaches, FaSAS introduces a progressive feedback adjustment mechanism that automatically selects a more appropriate verifier according to the failed one. This further reduces the difficulty and effort required for manually adjusting verifiers.

## A.2 EXPERIMENT

**Verifier suites.** For evaluation, we choose a set of verifiers from the SV-COMP website[3], including 2LS (Malík et al., 2023), CBMC (Kroening & Tautschnig, 2014), CPAchecker (Baier et al., 2024), DepthK (Rocha et al., 2017), ESBMC-kind (Menezes et al., 2024), ESBMC-incr (Menezes et al., 2024), Symbiotic (Jonáš et al., 2024), UAutomizer (Heizmann et al., 2015), UKojak (Nutz et al., 2015), and UTaipan (Dietsch et al., 2023). These verifiers have more or less participated in the four

---

[3]https://sv-comp.sosy-lab.org/2024/systems.php

verification tracks mentioned previously. Their supported techniques and algorithms can be found in the competition report(Beyer, 2024).

**Configuration.** When training the algorithm suggestion models and the verifier ranking model, we used the Adam training optimizer for each model over 20 rounds and set the learning rate to 1e-3. For `FaSAS`, we performed hyper-parameter tuning to search the threshold $\alpha$ in Eq. 5 and the adjustment coefficient $\beta$ in Eq. 7. We found $\alpha = 0.8$ and $\beta = 0.1$ to be optimal. For the two machine learning-based baselines mentioned in Sect. 4, we directly use their open-source implementation with the default configuration. To ensure the fairness of comparison results, we retrained all the models used in these baseline approaches with the same datasets as ours.

**Experiment Environment.** All the experiments are performed on a server running the Ubuntu 20.04 LTS system with a 2.2 GHz CPU and 503 GB RAM. We set the verification time bound and memory limitation of a verifier to 15 minutes and 15 GB, respectively.

**Threats to Validity.** While our experimental results demonstrate the effectiveness of `FaSAS`, several threats to validity must be acknowledged to provide a comprehensive understanding of the approach's limitations and generalizability. These threats mainly arise from dataset selection the semantic-preserving transformations. First, the datasets used for training and evaluation may contain biases or limitations. To address this concern, we made sure to randomly select diverse training data that covers a wide range of function calling patterns with different characteristics and program behaviors. Second, `FaSAS` relies on semantic-preserving transformations to enhance the robustness of the prediction model. However, ensuring that transformations are truly semantic-preserving can be challenging, and any deviations could impact the model's performance. Rigorous validation of these transformations is necessary to maintain the integrity of the approach.

