# OpenReview forum: "FaSAS: A Feedback-Augmented Stepwise Algorithm Selection for Software Verification"
_ICLR.cc/2026/Conference — Submitted to ICLR 2026_

### Official Review · Reviewer_y7c6 · 2025-10-31

**Soundness:** 3
**Presentation:** 3
**Contribution:** 2
**Rating:** 2
**Confidence:** 4

**Summary:**

FASAS is an algorithm selection method for selecting software verification tools for a given instance of program and property specification. FASAS operates in two stages: the first model generates positive/negative scores on different verification algorithms based on their applicability to the given instance or otherwise; the second model concatenates these score vectors and predicts the underlying verification tool to use. If the tool fails in verification, the scores are adjusted and another prediction is made for the next best tool. The paper presents evaluation results on sv-comp benchmark of 15K verification problems and compares against other algorithm selection approaches, showing better performance.

**Strengths:**

- Software verification is an important topic and using ML techniques to improve effectiveness of verification tools is a promising direction.
- The paper is nicely written and motivates the problem clearly. The main design based on code property graphs, GNNs and scoring, and the training details are explained in details.
- The experimental results on sv-comp on 15K problems and 20 verifications show that FASAS gives improvements over the baselines. The paper also argues about the benefit of decoupling the prediction in two stages of predicting algorithms and then the tools.

**Weaknesses:**

- The scope of the paper is narrow and is mostly of interest to software verification community. The ML contribution is slim. The paper is well-executed but the novelty is not significant.
- The paper does not evaluate state-of-the-art code embedding methods and LLMs (e.g., using few-shot prompts). It is therefore unclear why one needs to train a specialized model.
- The paper does not evaluate their approach on real-world code repositories or on multiple program languages.
- Remark: The paper cites recent work on algorithm selection for software verification appropriately. However, this problem has been explored much earlier. The earliest formulation of algorithm selection for verification (and subsequent progress) can also be cited:
Varun Tulsian, Aditya Kanade, Rahul Kumar, Akash Lal, and Aditya V. Nori.
Mux: Algorithm selection for software model checkers. 2014.

**Questions:**

- How does your method compare against pre-trained embedding models and LLMs?
- Why not train classifiers on top of pre-trained embedding models instead of training a GNN from scratch?
- Any evidence that FASAS works well on real-world repositories and multiple programming languages?

---

### Official Review · Reviewer_pVNw · 2025-10-31

**Soundness:** 3
**Presentation:** 3
**Contribution:** 2
**Rating:** 4
**Confidence:** 3

**Summary:**

This paper proposes FaSAS, an automated algorithm selection framework for software verification. The core idea is to model source code via Code Property Graphs (CPG) and employ Graph Neural Networks (GNNs) to predict potentially applicable verification algorithms, followed by a verifier-ranking process with iterative feedback refinement. The method aims to avoid reliance on high-quality labeled datasets while achieving scalability when new verifiers are introduced. Experimental results on SV-COMP benchmarks show higher accuracy and robustness compared to existing algorithm selectors such as Graves and CST.

**Strengths:**

1. The problem of algorithm selection for software verification is timely and practically relevant.

2. The decomposition of the selection process into multiple stages (algorithm suggestion, verifier matching, and feedback adjustment) is technically interesting.

3. The use of CPG-based representation and GNN modeling aligns well with the structural nature of program analysis.

4. The experimental evaluation is extensive and covers robustness and scalability studies.

**Weaknesses:**

1. The core idea is primarily a meta-selection framework — the model itself does not conduct verification but selects among existing tools. This limits its novelty from the algorithmic verification standpoint.

2. While the reported accuracy improvements are moderate (around +2–3% over Graves), the overall gain may not justify the additional model complexity.

3. The pipeline involves several neural components (two-stage prediction and feedback adjustment), which raises concerns about computational and implementation overhead in real-world deployment.

4. It would strengthen the paper to include more analysis on resource trade-offs (e.g., runtime cost vs. verification gain) and on how FaSAS behaves when faced with previously unseen verifier architectures.

**Questions:**

See weakness

---

### Official Review · Reviewer_ULtD · 2025-10-31

**Soundness:** 2
**Presentation:** 2
**Contribution:** 2
**Rating:** 6
**Confidence:** 3

**Summary:**

This paper presents a predictive model for the automated selection of appropriate verification tools. The model uses graph neural networks to embed the code property graph of the semantic-preserving transformed verification program. Results on examples from SV-Comp show advantage over previous approaches, such as Graves.

**Strengths:**

The paper contributes a new method for algorithm selection in software verification domain.

The paper presents compelling evidence taht the proposed method has merits as compared to previous approaches.

**Weaknesses:**

I found the work interesting but the presentation is unclear (please see below).

The paper may be of limited interest to machine learning community.

**Questions:**

The authors claim that the novelty of the method is that it does not need high-quality labeled sample (line 082). Yet it appears that the method does use negative and positive samples as explained in section 3.2.1. Can you please clarify?

I could not understand if the two models trained in step 2 need to be trained specifically wrt a program input? Can you please clarify?

There is no discussion about the overhead introduced by FaSAS. How does it compare with Graves?

---

### Official Review · Reviewer_t8ac · 2025-11-10

**Soundness:** 3
**Presentation:** 2
**Contribution:** 2
**Rating:** 2
**Confidence:** 3

**Summary:**

This paper proposes FaSAS, an automated algorithm selection framework for software verification. FaSAS avoids the need for algorithm-labeled datasets by leveraging a stepwise prediction pipeline: it first predicts potentially suitable verification algorithms from code property graphs (CPGs) using GNN embeddings, then matches them to verifiers. A progressive feedback loop refines the predictions based on failed verifications.
Instead of requiring expert-labeled algorithm annotations, FaSAS infers algorithm applicability from verifier performance. The assumption is: if a verifier succeeds on program P, the algorithms it implements are likely appropriate for P. When a verifier fails, FaSAS applies iterative refinement by adapting the scores of each algorithm.  Experiments on SV-COMP 2024 (20 verifiers, 15k C programs) show FaSAS achieves better  performances than SOTA with better robustness and scalability.

**Strengths:**

* The indirect labeling mechanism is creative, deriving algorithm suggestions from verifier performance without manual annotation. This tackles a real challenge in the verification domain.
* No need for labeled datasets — uses indirect labeling via verifier performance, which is elegant and scalable.
*  Extensive experiments including baselines, ablation studies (program representations, stepwise vs E2E, feedback rounds), robustness testing with code perturbations, and scalability analysis.
* Strong empirical results with clear improvements over SOTA (Graves, CST).

**Weaknesses:**

* I think that the stepwise procedure is what drives performance. Table 4 reveals that program representation choice barely matters—all representations achieve >91% accuracy. This means the stepwise learning procedure, not the CPG representation, is the real contribution. But this procedure is entirely ad hoc and not well formalized. Every component of this procedure lacks principled justification : why a scoring function made by hand and not learned (eq 2) ? Why an heuristic feedback mechanism and not optimized or learned ? For a paper targeting publication at an ML conference, this is very problematic.
* Some important details of the experimental setup are not very clear : The matrix C  in section 3.3 is restricted to training data or computed on the full dataset ?   If C is computed on the full dataset, this is a direct data leakage. The impressive results, especially with feedback, may be inflated by this leakage.
* The indirect labels used for training are derived from verifier performance on the same benchmark. The high accuracy may partially reflect the model learning to reproduce the training label generation process rather than generalizing to truly novel verification task. A better cross-domain (and cross-language) evaluation  should be performed.
* Writing is somewhat dense,  which could be clearer for a general ML audience. Many program verification concepts are referenced without real explanation, which is concerning for readers unfamiliar with the domain.
* Section A.1 with related work in appendix should be in main paper. T

**Questions:**

* Is matrix C computed only on training data? If not, why is this not leakage?
*  Does dual model (S+, S-) outperform single model ?
*  Can you test on any non-SV-COMP benchmark? (OOD performance)
* Why the specific Eq. 2 formulation? Have you tried learned scoring functions? or to learn this function ?

---

### Meta-Review · Area_Chair_hFbo · 2026-01-07

**Summary:**

The paper proposes FaSAS, a GNN-based algorithm selector for software verification. The consensus for rejection is driven by the method's limited novelty, as it relies heavily on ad-hoc heuristics and manually crafted scoring functions. Moreover, the scope of this paper is narrow and is mostly of interest to the software verification community, rather than the ML community.

**Reviewer Concerns:**

The author did not submit any discussion or rebuttal. So all problems are still unresolved.

**Reviewer Scores:**

The author did not submit any discussion or rebuttal, and it is not likely that any reviewer would have changed their scores.

---

### Decision · Program_Chairs · 2026-01-26

Reject